# Putting BPR in a Box: Bounding the Score Space in Bayesian Personalized Ranking

## Abstract

Bayesian Personalized Ranking (BPR) has been widely adopted for recommendation by optimizing pairwise score objectives. However, existing BPR-based methods typically focus on enlarging pairwise score differences, which often leads to excessive separation or clustering of pairwise data—an issue that can result in suboptimal performance. In this work, we propose BoxBPR, a novel framework that introduces explicit box constraints into the pairwise score space of BPR. Specifically, we present the motivation and formulation of both lower and upper bounds, derive a simple yet effective constraint based on the relationships among pairwise data, and directly integrate it into the BPR objective. We then introduce the optimization criteria of BoxBPR and describe the corresponding training process. From both lower- and upper-bound perspectives, we demonstrate that BoxBPR establishes a stronger connection to key top-$K$ evaluation metrics than BPR in recommendation tasks. Extensive experiments on three real-world datasets validate the effectiveness of BoxBPR, and comprehensive analyses further highlight the critical role of lower- and upper-bound constraints in BoxBPR.

## 1 Introduction

Recommender systems aim to model user preferences from historical interaction data to generate personalized ranking recommendations Zangerle & Bauer (2022); Yu et al. (2023); Lin et al. (2025). In practice, implicit feedback—such as clicks, views, or purchases—has become the most widely used source of historical interactions due to its simplicity and ease of collection. Personalized ranking from implicit feedback presents unique challenges: it provides only positive observations, while unobserved interactions are treated as ambiguous negatives Zhang et al. (2025); Chen et al. (2021); Zhu et al. (2024). This setting has motivated the development of ranking-based learning methods, among which Bayesian Personalized Ranking (BPR) Rendle et al. (2009) has emerged as one of the most influential approaches for implicit feedback. BPR seeks to maximize the score difference between positive and negative items. Over the past decade, numerous BPR variants have been proposed, either by incorporating additional signals (e.g., semantic information Wang et al. (2021) or diverse user preferences Wang et al. (2019); Pan et al. (2013a)) or by designing more effective sampling strategies (e.g., hard negative mining Ma et al. (2024); Lai et al. (2024); Yang et al. (2024)) to further enhance ranking performance.

Despite the rapid progress of BPR-based recommendation methods, the core optimization principle of BPR and its numerous variants remains largely unchanged: maximizing the score difference between positive and negative items Rendle et al. (2009); Shi et al. (2023); Zhu et al. (2024). In practice, these methods assign higher values to user–positive item pairs than to user–negative item pairs, with the optimization objective encouraging the score difference to be as large as possible. While this approach has proven effective in improving ranking performance, it imposes no explicit constraints on the absolute magnitude of the score difference. The absence of such bounds can lead to suboptimal performance in personalized ranking tasks, as well as creating a mismatch between the optimization objective and the actual evaluation criteria. As illustrated in Fig. 1, models trained solely under the BPR optimization produce excessively large score differences for some positive–negative pairs. Such over-separation can lead to situations where a few positive training items are assigned extremely high scores, while other relevant items—especially those appearing only in the test phase—are assigned disproportionately lower scores, thereby being ranked lower than their true relevance would suggest. On the other hand, the same model may also output overly clustered scores for some positive–negative

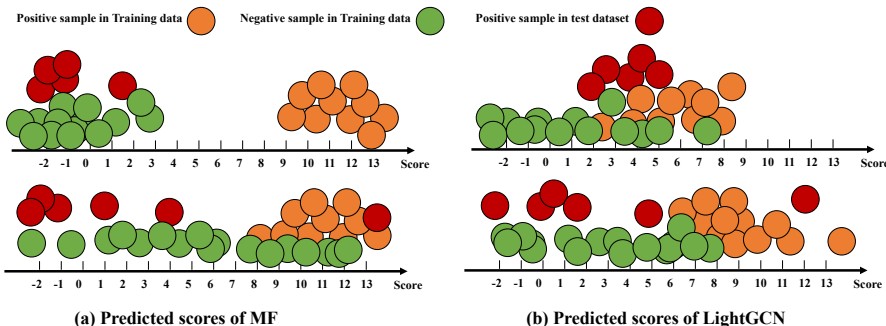

Figure 1: Illustration of the predicted scores for a user by BPR-based methods (MF and LightGCN) on the Yelp dataset. The results highlight two typical issues: (1) excessive separation, where maximizing pairwise score differences pushes positive and negative samples in the training data too far apart, leading to positive samples in the test data being ranked closer to negative items; and (2) score clustering, where the strong emphasis on separation compresses other pairwise scores, resulting in overlapping distributions of positive and negative samples in the training data and thus reducing their distinguishability.

pairs, making it difficult to distinguish between positive and negative items. These issues stem from the lack of explicit lower and upper bounds on pairwise score differences, which causes uncontrolled score scaling and numerical instability.

Moreover, BPR and its variants primarily optimize the AUC (Area Under the Curve), which measures the ability to distinguish positive from negative items across all possible pairs Rendle et al. (2009); Shi et al. (2023). Although useful, this objective treats all pairs equally, regardless of whether the items appear at the top or bottom of the ranking list. As a result, the optimization process is not directly aligned with improving key top-$K$ performance—where only the highest-ranked items matter in practical recommendation scenarios Rendle et al. (2009); Chen et al. (2020); Anand & Maurya (2025). Without explicit constraints, the model may push score differences to extreme values in order to improve AUC, leading to excessive separation as illustrated in Fig. 1, even if this provides little or no benefit for the top-$K$ positions. Such overemphasis on separating arbitrary pairs can distort the score scale and ultimately degrade ranking metrics.

To this end, we propose BoxBPR, a novel extension of BPR that incorporates explicit lower- and upper-bound constraints into the score space. The key idea is to treat predicted scores as variables confined within a "box" defined by well-motivated bounds, ensuring that the gap between positive and negative items is optimized while keeping scores in a reasonable range. Specifically, we derive a simple yet effective formulation of bounds from the relative relationship between positive and negative samples and incorporate them directly into the BPR optimization structure. We then present the corresponding optimization criteria and training algorithm, which can be seamlessly applied to existing BPR-based models. From a theoretical analysis, we show that BoxBPR establishes a stronger connection between pairwise optimization and top-K evaluation metrics compared to standard BPR. This connection arises because the bounded formulation better controls score magnitudes, making the learned ranking function more aligned with the objectives used in ranking evaluation. Empirically, extensive experiments on three real-world benchmark datasets demonstrate the effectiveness of BoxBPR, with consistent improvements over strong BPR-based baselines. Furthermore, comprehensive ablation and analytical studies reveal the complementary roles of lower and upper bounds, offering new insights into the role of score constraints in BPR-based optimization.

## 2 RELATED WORK

Personalized ranking has been extensively studied in recommender systems, with Bayesian Personalized Ranking (BPR) Rendle et al. (2012) emerging as one of the most widely adopted methods. Subsequent research has largely extended BPR along two main directions: (i) enhancing preference modeling by incorporating richer information signals, and (ii) improving training efficiency through more informative negative sampling strategies. The first line of work enriches BPR with additional

signals. For instance, GBPR Pan et al. (2013a) introduces group-level preferences, CoFiSet Pan et al. (2014) models item-set choices, RankMBPR Pan et al. (2013b) considers heterogeneous user attitudes, and ABPR Ouyang et al. (2014) extends this idea to groups. M-BPR Wang et al. (2019) generalizes BPR by incorporating multiple preference types. Beyond the pairwise assumption, similarity-based methods such as MSBPR Zeng et al. (2023), SMSBPR Zeng et al. (2025), and FSBPR Zheng & Wang (2024) integrate similarity constraints or embeddings. Other refinements include symmetric pairwise modeling Pan et al. (2023), higher-order relations Pan et al. (2014); Ni et al. (2022), and topic-aware extensions such as TDD-BPR Wang et al. (2021). The second line of work focuses on negative sampling Ma et al. (2024); Lai et al. (2024); Yang et al. (2024). Adaptive strategies Zhang et al. (2013); Yang et al. (2024) adjust sampling according to model states, such as DNS Zhang et al. (2013), SRNS Ding et al. (2020), and DENS Zhang et al. (2023). Adversarial methods (e.g., IRGAN, KBGAN, ADVIR Wang et al. (2017); Cai & Wang (2018); Fan et al. (2022)) and recent designs such as MixGCF Zhao et al. (2021) and AHNS Lai et al. (2024) further synthesize or adaptively select harder negatives. While these approaches enhance BPR's ranking performance, they often overlook a key limitation: the unbounded expansion of the score space in BPR may lead to predicted scores being excessively separated or overly clustered, ultimately resulting in suboptimal performance.

## 3  PRELIMINARY

In recommendation systems based on implicit feedback, the prediction objective is to estimate a user's preference score for each item and then rank items in descending order of these scores to generate the final recommendation list. Let $U$ denote the set of all users and $V$ the set of all items. The implicit feedback interaction matrix is defined as:

$$R = [r_{uv}]_{|U| \times |V|} \in \{0, 1\}, \tag{1}$$

where $r_{uv} = 1$ indicates that the interaction between user $u$ and item $v$ is observed (i.e., user $u$ has expressed positive implicit feedback toward $v$), and $r_{uv} = 0$ indicates negative feedback (i.e., user $u$ has not interacted with $v$, which may imply either disinterest or potential interest). For each user $u \in U$, we denote the set of items with positive feedback as:

$$V_u^+ := \{v \in V \mid (u, v) \in R\}, \tag{2}$$

and the set of items with negative feedback as

$$V_u^- := \{v \in V \mid (u, v) \notin R\}. \tag{3}$$

Given the interaction matrix $R$, the goal of recommendation is to predict scores $\hat{R} = [\hat{r}_{uv}]_{|U| \times |V|}$ that accurately reflect users' interests in items, thereby improving personalized ranking results. Our focus is on designing a more effective optimization objective, rather than constraining the architecture of the recommendation model, to enhance recommendation performance.

### 3.1  BPR OPTIMIZATION

In recommendation model with the Bayesian Personalized Ranking (BPR) Rendle et al. (2009); Chen et al. (2020); He et al. (2020), the learning objective is to optimize the pairwise ranking of items such that, for each user $u$, the predicted preference score for the positive item $i \in V_u^+$ is higher than that for the negative item $j \in V_u^-$. Formally, the BPR loss function is given by:

$$L_{\text{BPR}} = -\sum_{u \in U} \sum_{i \in V_u^+} \sum_{j \in V_u^-} log\, \sigma(\hat{r}_{ui} - \hat{r}_{uj}), \tag{4}$$

where $\sigma(\cdot)$ is the logistic sigmoid function, and $\hat{r}_{ui}$ denotes the predicted preference score of user $u$ for item $i$. The core idea of BPR is to model the preference structure $i >_u j$, meaning that user $u$ is more likely to prefer item $i$ over item $j$.

It is important to note that BPR and its variants only require the score of a positive item to be higher than that of a negative item, without explicitly constraining the magnitude of the score range. However, unconstrained optimization may cause the predicted scores to become either excessively separated or undesirably clustered, leading to suboptimal ranking performance. To address this issue, we focus on defining an optimal range for the score difference between positive and negative items.

## 4 METHODOLOGY

In this work, we propose BoxBPR, a novel optimization framework that introduces box constraints into BPR to regulate the score space and better align the optimization process with ranking metrics.

### 4.1 CONSTRUCTION OF BOUNDS

To ensure that the proposed box constraints are both practical and effective, we highlight two key design principles that guide their formulation.

**Pair-specific constraints.** The proposed bounds are defined at the level of specific user–item pairs. For different pairs, the feasible score range can naturally vary, since the predicted scores depend on the characteristics of both the user and the item. This design acknowledges that not all interactions require the same margin to be meaningful, thereby allowing the model to flexibly adapt the margin according to the context of each pair.

**Dynamic adjustment during training.** The bounds are not fixed but evolve dynamically during training. As the recommendation model is optimized under the BPR framework, its parameters and learned embeddings continuously update, which in turn alters the predicted scores. Consequently, the feasible upper and lower bounds of the score difference must also adjust throughout training. This adaptive mechanism ensures that the margin constraints remain effective and aligned with the evolving embedding space.

Guided by the above design principles, we now formalize the lower and upper bound constraints on the score between positive and negative items. Specifically, the constraints are defined as:

$$\hat{r}_{ui} - \hat{r}_{uj} - \alpha\,\hat{r}_{ui} > 0, \tag{5}$$
$$\hat{r}_{ui} - \hat{r}_{uj} - \beta\,\hat{r}_{uj} < 0, \tag{6}$$
$$\beta\,\hat{r}_{uj} > \hat{r}_{ui} - \hat{r}_{uj} > \alpha\,\hat{r}_{ui}, \tag{7}$$
$$\tag{8}$$

where $\alpha < 1$, $\beta < 1$. $\hat{r}ui$ and $\hat{r}uj$ denote the predicted scores produced by an arbitrary recommendation model. These constraints jointly ensure that the score difference remains within a reasonable range that is adaptively determined by both scores. The lower bound, controlled by $\alpha\,\hat{r}_{ui}$, prevents the score difference from being too small, ensuring that the model sufficiently favors the positive item. The upper bound, controlled by $\beta\,\hat{r}_{uj}$, prevents the score difference from becoming excessively large, avoiding the situation where the positive score grows without bound or the negative score becomes unrealistically small.

This dual-constraint optimization acts as a balancing mechanism: the inner objective encourages increasing $\hat{r}_{ui}$ and decreasing $\hat{r}_{uj}$ to maximize the score difference, while the outer bounds limit this gap to a controlled range. As a result, the learned ranking satisfies $i >_u j$ for each user-item pair while maintaining stability and preventing overconfident or extreme predictions.

### 4.2 OPTIMIZATION CRITERIA OF BOXBPR

In BoxBPR, the idea of simultaneously imposing both upper and lower bounds on the BPR optimization is formalized by maximizing two posterior probability terms. Let $\Theta$ denote the parameter vector of an arbitrary recommendation model. The lower-bound constraint ensures that the score difference between a positive and a negative item is sufficiently large, while the upper-bound constraint prevents the score difference from becoming excessively large. These constraints can be expressed in a Bayesian form as:

$$p(\Theta|(\hat{r}_{ui} - \hat{r}_{uj}) > \alpha\,\hat{r}_{ui}) \propto p((\hat{r}_{ui} - \hat{r}_{uj}) > \alpha\,\hat{r}_{ui}|\Theta)p(\Theta), \tag{9}$$
$$p(\Theta|(\hat{r}_{ui} - \hat{r}_{uj}) < \beta\,\hat{r}_{uj}) \propto p((\hat{r}_{ui} - \hat{r}_{uj}) < \beta\,\hat{r}_{uj}|\Theta)p(\Theta), \tag{10}$$

where $>$ denotes the desired personalized ranking relation that is consistent with the user's latent preference structure.

Following the same assumption as in BPR, we treat the ordering of each pair—comprising one positive and one negative sample—as independent from the ordering of any other pair in the dataset.

Under this independence assumption, the likelihood term $p(* \mid \Theta)$ can be factorized into the product of individual pairwise likelihoods and subsequently aggregated over all users $u \in U$:

$$\prod_{u \in U} p((\hat{r}_{ui} - \hat{r}_{uj}) > \alpha\,\hat{r}_{ui}|\Theta) = \prod_{u \in U, i \in V_u^+, j \in V_u^-} p((\hat{r}_{ui} - \hat{r}_{uj}) > \alpha\,\hat{r}_{ui}|\Theta), \qquad (11)$$

$$\prod_{u \in U} p((\hat{r}_{ui} - \hat{r}_{uj}) < \beta\,\hat{r}_{uj}|\Theta) = \prod_{u \in U, i \in V_u^+, j \in V_u^-} p((\hat{r}_{ui} - \hat{r}_{uj}) < \beta\,\hat{r}_{uj}|\Theta), \qquad (12)$$

$$(13)$$

To ensure the personalized ranking order, we follow the same definition of individual pairwise probabilities as in BPR, where the probability reflects the likelihood that a user prefers a positive sample over a negative sample. The key difference in our formulation is that we impose both upper and lower preference bounds. Specifically, the BoxBPR is defined as:

$$p((\hat{r}_{ui} - \hat{r}_{uj}) > \alpha\,\hat{r}_{ui}|\Theta) = p((1 - \alpha)\hat{r}_{ui} > \hat{r}_{uj})|\Theta), \qquad (14)$$
$$:= \sigma((1 - \alpha)\hat{r}_{ui} - \hat{r}_{uj})), \qquad (15)$$
$$p((\hat{r}_{ui} - \hat{r}_{uj}) < \beta\,\hat{r}_{uj}|\Theta) = p((1 + \beta)\hat{r}_{uj}) > \hat{r}_{ui}|\Theta), \qquad (16)$$
$$:= \sigma((1 + \beta)\hat{r}_{uj} - \hat{r}_{ui})) \qquad (17)$$

where $\sigma(\cdot)$ is the logistic sigmoid function, and $\alpha < 1, \beta < 1$ are hyperparameters controlling the lower and upper margin ratios, respectively. The predicted values $\hat{r}_{ui}$ and $\hat{r}_{uj}$ are computed based on the model parameters $\Theta$, which capture the latent relationships among user $u$, positive item $i \in V_u^+$, and negative item $j \in V_u^-$. For notational simplicity, we omit $\Theta$ from $\hat{r}_{ui}$ and $\hat{r}_{uj}$.

Building on the above likelihood definitions, we adopt the maximum a posteriori estimator to derive our generic optimization criterion for personalized ranking. Under the independence assumption, the joint posterior of the lower-bound and upper-bound constraints can be expressed as:

$$
\begin{aligned}
\text{BoxBPR} :=& log\, p(\Theta|(\hat{r}_{ui} - \hat{r}_{uj}) < \beta\,\hat{r}_{uj}) + log\, p(\Theta|(\hat{r}_{ui} - \hat{r}_{uj}) > \alpha\,\hat{r}_{ui}), \\
=& log\, p((\hat{r}_{ui} - \hat{r}_{uj}) < \beta\,\hat{r}_{uj}|\Theta)p(\Theta) + log\, p((\hat{r}_{ui} - \hat{r}_{uj}) > \alpha\,\hat{r}_{ui}|\Theta)p(\Theta), \\
=& log \prod_{u \in U, i \in V_u^+, j \in V_u^-} \sigma((1 - \alpha)\hat{r}_{ui} - \hat{r}_{uj}))p(\Theta) + \\
& log \prod_{u \in U, i \in V_u^+, j \in V_u^-} \sigma((1 + \beta)\hat{r}_{uj} - \hat{r}_{ui}))p(\Theta), \\
=& \sum_{u \in U, i \in V_u^+, j \in V_u^-} log\, \sigma((1 - \alpha)\hat{r}_{ui} - \hat{r}_{uj})) + \\
& \sum_{u \in U, i \in V_u^+, j \in V_u^-} log\, \sigma((1 + \beta)\hat{r}_{uj} - \hat{r}_{ui})) + 2 * log\, p(\Theta), \\
=& \sum_{u \in U, i \in V_u^+, j \in V_u^-} log\, \sigma((1 - \alpha)\hat{r}_{ui} - \hat{r}_{uj})) + \\
& \sum_{u \in U, i \in V_u^+, j \in V_u^-} log\, \sigma((1 + \beta)\hat{r}_{uj} - \hat{r}_{ui})) - \gamma_\Theta\,||\Theta||^2.
\end{aligned}
\qquad (18)
$$

Here, $\sigma(\cdot)$ is the logistic sigmoid function, and $\gamma_\Theta$ is the regularization coefficient. This formulation unifies the BPR objective with bound-aware score difference constraints, allowing the model to maintain the personalized ranking order while avoiding overly small or excessively large score difference. Since both bounds are derived directly from the current pairwise data without introducing additional variables, BoxBPR requires only minimal changes to the standard BPR formulation and can be seamlessly integrated into BPR-based recommendation models.

## 4.3 ANALYSIS OF BOXBPR

In this subsection, we provide a mathematical derivation showing why adding a *box constraint*

$$\alpha\,\hat{r}_{ui} \le \Delta_{uij} \le \beta\,\hat{r}_{uj}, \qquad \Delta_{uij} = \hat{r}_{ui} - \hat{r}_{uj}, \quad \alpha < 1,\ \beta < 1$$

enhances *Top-K ranking performance* (HRK, NDCGK).

Let user $u$ have candidate set $I_u = \{i\} \cup V_u^-$, where $i$ is the positive item and $V_u^-$ is the set of negatives. Let the negative scores be sorted in descending order:

$$s_u^{(1)} \geq s_u^{(2)} \geq \cdots \geq s_u^{(|V_u^-|)}, \quad s_u^{(k)} := \text{the } k\text{-th largest among } \{\hat{r}_{uj} : j \in V_u^-\}.$$

The rank of the positive item $i$ is

$$\mathrm{rank}_u(i) = 1 + \left|\{j \in V_u^- : \hat{r}_{uj} \geq \hat{r}_{ui}\}\right|,$$

and the Top-K condition can be written as

$$\mathrm{rank}_u(i) \leq K \quad \Longleftrightarrow \quad \hat{r}_{ui} > s_u^{(K)}.$$

**1) Lower Bound $\Delta_{uij} \geq \alpha\,\hat{r}_{ui}$ as a Sufficient Condition**

Let $j^\star$ denote the negative item at rank $K$, i.e., $\hat{r}_{uj^\star} = s_u^{(K)}$. If the lower bound holds:

$$\Delta_{uij^\star} \geq \alpha\,\hat{r}_{ui} \quad \Rightarrow \quad \hat{r}_{ui} - s_u^{(K)} \geq \alpha\,\hat{r}_{ui} \quad \Rightarrow \quad \hat{r}_{ui} \geq \frac{s_u^{(K)}}{1-\alpha} > s_u^{(K)},$$

which ensures $\mathrm{rank}_u(i) \leq K$. Enforcing a lower bound on the pairwise difference for the $K$-th hard negative provides a *sufficient condition* for achieving better Top-K metrics.

**2) Upper Bound $\Delta_{uij} \leq \beta\hat{r}_{uj}$ and Gradient Stabilization**

The standard BPR loss $\ell_{\mathrm{bpr}}(\Delta_{uij}) = -log\sigma(\Delta_{uij})$ has derivative

$$\ell'_{\mathrm{bpr}}(\Delta_{uij}) = \sigma(\Delta_{uij}) - 1 \in (-1, 0), \quad |\ell'_{\mathrm{bpr}}(\Delta_{uij})| = \sigma(-\Delta_{uij}),$$

which vanishes as $\Delta_{uij} \to \infty$, leading to gradient depletion.

Adding an upper-bound penalty

$$\ell_U(\Delta_{uij}, \hat{r}_{uj}) = \max(0, \Delta_{uij} - \beta\hat{r}_{uj})$$

yields gradient

$$\frac{\partial}{\partial \Delta_{uij}}(\ell_{\mathrm{bpr}} + \ell_U) = \begin{cases} \sigma(\Delta_{uij}) - 1, & \Delta_{uij} \leq \beta\hat{r}_{uj}, \\ \sigma(\Delta_{uij}) - 1 + (\Delta_{uij} - \beta\hat{r}_{uj}), & \Delta_{uij} > \beta\hat{r}_{uj}. \end{cases}$$

Hence, maintains gradients in the effective range, preventing vanishing updates for hard negatives near the Top-K boundary. Limits relative score inflation: $\hat{r}_{ui} \leq (1+\beta)\hat{r}_{uj}$, preserving the scale of top-ranking items and stabilizing NDCG. Specifically, From the upper-bound constraint $\hat{r}_{ui} \leq (1+\beta)\,\hat{r}_{uj}$, we can see that the predicted score of a positive item $\hat{r}_{ui}$ is bounded relative to the score of the corresponding negative item $\hat{r}_{uj}$. In particular, when $j$ is selected as a hard negative sample, this inequality constrains $\hat{r}_{ui}$ to lie within the scale of the "top" negative items.

Imposing a *bounded pairwise constraint* $\alpha\hat{r}_{ui} \leq \Delta \leq \beta\hat{r}_{uj}$. Ensures the positive item exceeds the top-K negatives with high probability. Maintains effective gradients on hard negatives near the Top-K boundary. Prevents score inflation, stabilizing relative ordering within Top-K. These effects collectively explain the improvement in HR@K and NDCG@K.

## 5 EXPERIMENTS

### 5.1 EXPERIMENTS SETUP

**Datasets.** We select three public and widely used benchmark datasets to evaluate our proposed BoxBPR model, namely Yelp, Amazon-kindle, and Douban-Book Yu et al. (2021); Chen et al. (2025);

Zhao et al. (2016). These datasets cover different recommendation scenarios and exhibit varying levels of interaction sparsity. Following common practices in recommendation research He et al. (2020); Ding et al. (2020); Chen et al. (2020), we adopt the 5-core filtering strategy to ensure data quality, and split each dataset into training, validation, and test sets with a ratio of 7:1:2. The detailed statistics of the datasets are summarized in Appendix A.2.

**Baselines.** Our experiments include two sets of baselines: one consists of recommendation model backbones, and the other comprises variants of BPR. The first group of baselines includes models optimized only with the BPR loss (e.g., MF, LightGCN He et al. (2020)) and models optimized with multiple loss functions (e.g., SimGCL Ding et al. (2020), NCL Lin et al. (2022), DCCF Ren et al. (2023), BIGCF Zhang et al. (2024), NLGCL Xu et al. (2025)). The second group of baselines includes enhanced BPR with more information(e.g., DNS Zhang et al. (2013), SRNS Ding et al. (2020), MixGCF Zhao et al. (2021) and AHNS Lai et al. (2024)).

**Evaluation Metrics and Implementation Details.** We evaluate the recommendation performance using the commonly used top-$K$ ranking metrics HR@K and NDCG@K, with $K = 20$ He et al. (2020); Chen et al. (2020; 2021). Following widely adopted practice Chen et al. (2020; 2021); Ni et al. (2022), we adopt the full-ranking evaluation protocol, where predictions are ranked against the entire item set. Since the proposed BoxBPR improves only the BPR optimization objective, we keep all other components identical to the original recommendation backbone. In addition, we tune the parameters consistently to ensure a fair comparison.

Table 1: Overall perfermance of BoxBPR on three datasets.

| Model | Yelp | | Amazon-kindle | | Douban-Book | |
|---|---|---|---|---|---|---|
| | HR@20 | NDCG@20 | HR@20 | NDCG@20 | HR@20 | NDCG@20 |
| MF | 0.04320 | 0.03941 | 0.11626 | 0.09131 | 0.09374 | 0.10313 |
| MF+BoxBPR | 0.04548 | 0.04213 | 0.1232 | 0.09608 | 0.10122 | 0.11511 |
| LightGCN | 0.05158 | 0.04809 | 0.14954 | 0.11921 | 0.10914 | 0.12207 |
| LightGCN+BoxBPR | 0.05551 | 0.05142 | 0.15750 | 0.12640 | 0.11775 | 0.13042 |
| NCL | 0.05910 | 0.05499 | 0.14777 | 0.11378 | 0.12022 | 0.13963 |
| NCL+BoxBPR | 0.06112 | 0.05628 | 0.15141 | 0.11621 | 0.12420 | 0.14480 |
| SimGCL | 0.06392 | 0.05993 | 0.15489 | 0.11678 | 0.13254 | 0.14609 |
| SimGCL+BoxBPR | 0.06414 | 0.0615 | 0.15960 | 0.11978 | 0.13547 | 0.14980 |
| DCCF | 0.06089 | 0.05395 | 0.15595 | 0.11667 | 0.13982 | 0.14332 |
| DCCF+BoxBPR | 0.06239 | 0.0558 | 0.15975 | 0.12077 | 0.14450 | 0.14861 |
| BIGCF | 0.06295 | 0.05672 | 0.15687 | 0.11606 | 0.14270 | 0.14839 |
| BIGCF+BoxBPR | 0.06459 | 0.05860 | 0.16072 | 0.11969 | 0.14652 | 0.15302 |
| NLGCL | 0.06590 | 0.06155 | 0.16534 | 0.12367 | 0.14826 | 0.16636 |
| NLGCL+BoxBPR | 0.06793 | 0.06334 | 0.16958 | 0.12770 | 0.15258 | 0.16980 |

## 5.2 OVERALL COMPARISON

**1) Comparison on backbones optimized solely with the BPR loss.** We first evaluate BoxBPR on some backbones (MF and LightGCN) that are optimized only with the BPR loss. This setting allows us to directly examine how BoxBPR improves over standard BPR optimization. As shown in Table 1, BoxBPR consistently achieves more than 5% performance gains across all three datasets. Specifically, the average improvements reach 6.68%, 5.64%, and 8.58% on Yelp, Amazon-Kindle, and Douban-Book, respectively. These results demonstrate the effectiveness and stability of BoxBPR in diverse recommendation scenarios.

**2) Comparison on backbones optimized with BPR plus additional losses.** We further evaluate BoxBPR on advanced backbones (NCL, SimGCL, DCCF, BIGCF and NLGCL) that combine the BPR loss with other auxiliary objectives. This setting reflects more complex optimization scenarios and allows us to assess the generality of BoxBPR. From Table 1, we observe that BoxBPR consistently yields improvements, with an average gain of 2.78% across all datasets, demonstrating its effectiveness. Compared with backbones optimized solely by the BoxBPR, the relative improvements are smaller, which can be explained by the fact that BoxBPR directly affects only a subset of objectives

Table 2: Performance of BoxBPR compared to BPR variants on all datasets

| Backbone | Method | Yelp | | Amazon-kindle | | Douban-Book | |
|---|---|---|---|---|---|---|---|
| | | HR@20 | NDCG@20 | HR@20 | NDCG@20 | HR@20 | NDCG@20 |
| MF | BPR | 0.04320 | 0.03941 | 0.11626 | 0.09131 | 0.09374 | 0.10313 |
| | DNS | 0.04370 | 0.04047 | 0.12144 | 0.09411 | 0.09397 | 0.10511 |
| | SRNS | 0.04419 | 0.03992 | 0.11950 | 0.09459 | 0.09421 | 0.10251 |
| | MixGCF | 0.04415 | 0.04102 | 0.11823 | 0.09496 | 0.09324 | 0.10684 |
| | AHNS | 0.04470 | 0.04135 | 0.12170 | 0.09511 | 0.09545 | 0.10939 |
| | BoxBPR | **0.04548** | **0.04213** | **0.12320** | **0.09608** | **0.10122** | **0.11511** |
| LightGCN | BPR | 0.05158 | 0.04809 | 0.14954 | 0.11921 | 0.10914 | 0.12207 |
| | DNS | 0.05313 | 0.04708 | 0.15128 | 0.12120 | 0.11115 | 0.12457 |
| | SRNS | 0.05202 | 0.04804 | 0.15494 | 0.12028 | 0.10745 | 0.12028 |
| | MixGCF | 0.05351 | 0.04973 | 0.15469 | 0.12113 | 0.10983 | 0.12603 |
| | AHNS | 0.05452 | 0.05054 | 0.15490 | 0.12410 | 0.11080 | 0.12751 |
| | BoxBPR | **0.05551** | **0.05142** | **0.15750** | **0.12640** | **0.11775** | **0.13042** |

in multi-loss optimization. Nevertheless, BoxBPR yields clear improvements on both HR and NDCG, highlighting its adaptability and generality across diverse backbone models.

**3) Comparison to BPR Variants** Consistent with the experimental setting of BPR variants Shi et al. (2023); Liu & Wang (2023); Lai et al. (2024), we also adopt MF and LightGCN as the CF backbones to validate the effectiveness of different BPR optimization methods. As shown in Table 2, our model achieves average improvements of 2.86% and 3.99% over the best baseline when using MF and LightGCN as backbones, respectively. In addition, across the three datasets, we obtain average gains of 2.29%, 1.44%, and 6.19%. These consistent improvements demonstrate the effectiveness of BoxBPR. This can be explained by the fact that existing BPR variants mainly enhance AUC optimization by incorporating additional information, whereas our method constrains the score space to directly optimize for top-$k$ metrics, which aligns more closely with the recommendation tasks and thus leads to better performance.

Table 3: Ablation study of BoxBPR.

| Model | Yelp | | Amazon-kindle | | Douban-Book | |
|---|---|---|---|---|---|---|
| | HR@20 | NDCG@20 | HR@20 | NDCG@20 | HR@20 | NDCG@20 |
| MF+BoxBPR | 0.04548 | 0.04213 | 0.12320 | 0.09608 | 0.10122 | 0.11511 |
| w/o low | 0.04453 | 0.04091 | 0.12023 | 0.09355 | 0.09822 | 0.11197 |
| w/o up | 0.04395 | 0.04061 | 0.11838 | 0.09290 | 0.09770 | 0.11222 |
| LightGCN+BoxBPR | 0.05551 | 0.05142 | 0.15750 | 0.12640 | 0.11775 | 0.13042 |
| w/o low | 0.05433 | 0.05041 | 0.15392 | 0.12337 | 0.115381 | 0.12788 |
| w/o up | 0.05437 | 0.05013 | 0.1533 | 0.12353 | 0.11497 | 0.12679 |

## 5.3 ABLATION STUDY

We conduct an ablation study to examine the effects of the lower- and upper-bound constraints in BoxBPR. In particular, we set $\alpha = 0$ in Equation 19 to remove the lower bound (w/o low) and set $\beta = 0$ to remove the upper bound (w/o up). For simplicity, we adopt MF and LightGCN as backbones, since these models are optimized solely with the BPR loss, making them more suitable for isolating the contributions of each constraint. Consistent trends are also observed across other backbone models. The results in Table 3 show that removing either the lower- or upper-bound constraint leads to performance drops of 2.43% and 2.99% on average, respectively. Nevertheless, both variants still outperform the standard BPR model, demonstrating the overall effectiveness of introducing bound constraints. These findings confirm that both bounds are crucial for BoxBPR, with each contributing significantly to its performance.

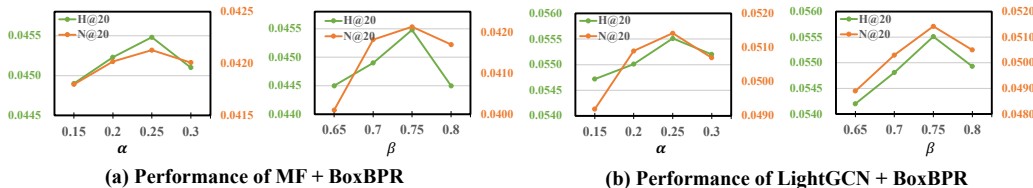

Figure 2: Sensitivity Analysis on Yelp w.r.t. $\alpha$ and $\beta$ in Eq. 18. "H@K" and "N@K" refer to "HR@K" and "NDCG@K", respectively.

## 5.4 SENSITIVITY ANALYSIS

We study the sensitivity of BoxBPR to the hyperparameters $\alpha$ and $\beta$ in Eq. 18, where $\alpha$ controls the strength of the lower bound and $\beta$ controls the strength of the upper bound. We conduct experiments using MF and LightGCN as the backbone models on the Yelp datasets. As shown in Fig. 2, different backbones exhibit the same optimal configurations of $\alpha$ and $\beta$, indicating that our model is stable and relatively insensitive to hyperparameter choices. Moreover, compared to $\alpha$, the parameter $\beta$ has a larger impact on performance, which is consistent with the ablation study where removing the upper-bound constraint leads to a larger performance drop. This suggests that constraining the upper bound plays a more critical role in achieving the best results for BoxBPR.

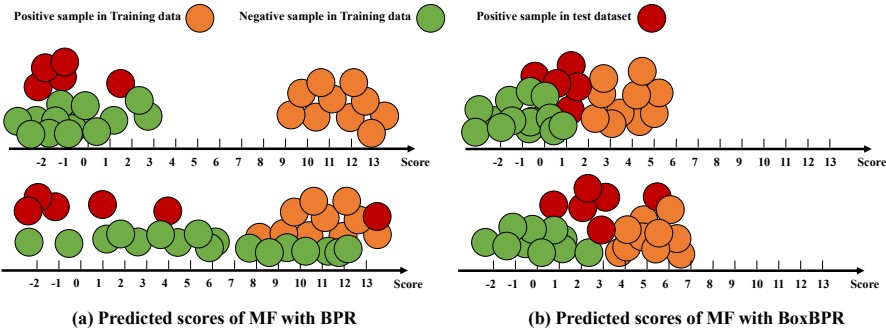

Figure 3: Comparison of results for the same user under BPR and BoxBPR with MF model.

## 5.5 CASE STUDY

In the case study, we compare the predicted score distributions on the Yelp dataset using the same MF backbone under BPR and BoxBPR optimization. Similar trends are consistently observed across other datasets and backbone models, with additional results provided in Appendix A.3. As shown in Fig. 3(b), BoxBPR effectively constrains the score space, leading to more distinguishable distributions between positive and negative samples. This avoids the excessive separation and clustering issues observed with BPR in Fig. 3(a), thereby yielding higher rankings for positive samples in the test set. Overall, the case study highlights both the effectiveness and the rationality of our proposed BoxBPR.

## 6 CONCLUSION

In this work, we highlight the limitations of existing BPR and its variants, which primarily incorporate additional information to better optimize the AUC objective but fail to directly target ranking metrics, often resulting in suboptimal performance. To address these issues, we design the BoxBPR optimization that explicitly bounds the score space for BPR-based optimization. In particular, we present the motivation and formulation of introducing both lower- and upper-bound constraints, and derive the corresponding optimization criteria and training details. Finally, extensive experiments on three real-world datasets validate the effectiveness of our model.

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

# A  Appendix

## A.1  The Use of Large Language Models

Yes, it helps refine writing by providing grammar checking and correction, thereby enhancing the overall quality of the text.

## A.2  Dataset

In our study, we conduct experiments on three publicly available datasets: Yelp, Amazon-Kindle, and Douban-Book. The Yelp dataset is sourced from the 2018 edition of the Yelp Challenge. The Amazon-Kindle dataset is extracted from Amazon reviews, focusing specifically on Kindle products. The Douban-Book dataset is collected from Douban, a popular book review website in China. Table A.2 summarizes the statistics of these three datasets.

Table 4: Statistics of the two datasets.

| Datasets | Users | Items | Interaction | Density |
|---|---|---|---|---|
| Yelp | 31,668 | 38,048 | 1,561,406 | 0.130% |
| Amazon-kindle | 138,333 | 98,572 | 1,909,965 | 0.014% |
| Douban-Book | 12,638 | 22,222 | 598,420 | 0.213% |

## A.3  Case Study

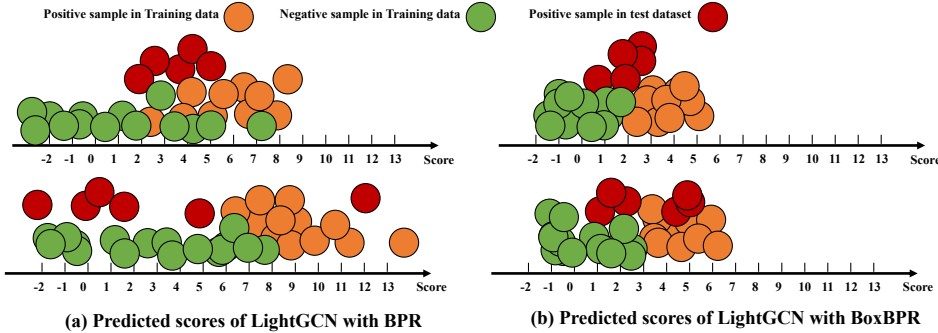

(a) Predicted scores of LightGCN with BPR    (b) Predicted scores of LightGCN with BoxBPR

Figure 4: Comparison of results for the same user under BPR optimization and BoxBPR optimization with LightGCN model.

In the case study, we analyze the predicted score distributions on the Yelp dataset using the same LightGCN backbone under both BPR and BoxBPR optimization. As illustrated in Fig. 4(b), BoxBPR effectively constrains the score space, resulting in more clearly separated distributions for positive and negative samples. Consistent patterns can also be observed in Fig. 3. Overall, the case study demonstrates both the effectiveness and the soundness of the proposed BoxBPR method.

