# OpenReview forum: "Putting BPR in a Box: Bounding the Score Space in Bayesian Personalized Ranking"
_ICLR.cc/2026/Conference — ICLR 2026 Conference Withdrawn Submission_

### Official Review · Reviewer_i1vb · 2025-10-17

**Soundness:** 1
**Presentation:** 2
**Contribution:** 1
**Rating:** 2
**Confidence:** 5

**Summary:**

This paper focuses on improving the standard Bayesian Personalized Ranking (BPR) loss in Recommender Systems. The authors identify a key limitation of BPR, which arises from the absence of score constraints on positive-negative score differences. To overcome this limitation, they introduce upper and lower bounds on these score differences, leading to a new loss function called BoxBPR. The authors argue that BoxBPR enhances the alignment between the optimization objective and the top-$K$ ranking metrics, thereby improving recommendation performance. Experiments on three datasets and seven backbones demonstrate the potential of BoxBPR.

**Strengths:**

- This paper studies the widely used BPR loss and identifies a key limitation.
- The proposed method is conceptually simple and easy to implement.
- The paper writes clearly and is easy to follow.

**Weaknesses:**

- The motivation is weak, as the score differences can be implicitly bounded by applying other scoring functions instead of the inner product.
- The proposed method lacks theoretical justification, epsecially on the connection between the objective and the top-$K$ ranking metrics.
- Some important baselines are missing in the experiments.

**Questions:**

- **Motivation.** The authors argue that the absence of score constraints in BPR leads to suboptimal performance, including the issue of excessive separation and score clustering. However, I have the following concerns:
  - **Concern 1**: These two issues are inherently contradictory, as the excessive separation makes the positive and negative scores diverge, while score clustering requires the positive and negative scores to be overlapped. So how can both issues exist simultaneously in BPR (e.g., the upper and lower parts of Figure 1)?
  - **Concern 2**: The score differences can be implicitly bounded by applying other scoring functions instead of the inner product, such as the sigmoid function and the cosine similarity. As these functions are widely used in recommender systems, why the unbounded issue still matters?
- **Theoretical Justification.** The authors claim that BoxBPR improves the alignment between the optimization objective and the top-$K$ ranking metrics, e.g., NDCG@$K$ and HR@$K$. However, I have the following concerns:
  - **Concern 1**: These claims are not theoretically justified by the extremely weak analysis in Section 4.3. The authors suggest that BoxBPR imposes a margin between the positive score and top-$K$ quantile of negative scores, thus improving the top-$K$ ranking metrics. However, this analysis requires all the lower bound constraints to be satisfied, which is equivalent to maximizing the objective. This assumption is too strong, as if the original BPR is maximized, all the positive items are ranked higher than the negative items, and thus the top-$K$ ranking metrics are also optimal. The authors need to provide more rigorous theoretical analysis to justify the effectiveness of the proposed method.
  - **Concern 2**: There are a lot of prior works on directly optimizing the top-$K$ ranking metrics, such as LambdaLoss@$K$ [R1], SONG@$K$ [R2], LLPAUC [R3], and SL@$K$ [R4]. This paper lacks comparison and discussion with these methods.
- **Experiments.** The experimental results are insufficient, since the above top-$K$ optimization methods are not included in comparisons.

**References:**

- [R1] On Optimizing Top-K Metrics for Neural Ranking Models. SIGIR '22.
- [R2] Large-scale Stochastic Optimization of NDCG Surrogates for Deep Learning with Provable Convergence. ICML '22.
- [R3] Lower-Left Partial AUC: An Effective and Efficient Optimization Metric for Recommendation. WWW '24.
- [R4] Breaking the Top-K Barrier: Advancing Top-K Ranking Metrics Optimization in Recommender Systems. KDD '25.

---

### Official Review · Reviewer_rdfv · 2025-10-29

**Soundness:** 2
**Presentation:** 3
**Contribution:** 1
**Rating:** 2
**Confidence:** 4

**Summary:**

The authors propose a method to overcome some limitations of the Bayesian Personalized Ranking (BPR) method, which constitutes the basis for numerousf implicit-feedback recommendation methods for over a decade. These limitations are claimed to be: (1)  an excessive score separation, where a few positives get very high scores, distorting the scale and (2) score overlapping, where scores for positives and negatives overlap due to uncontrolled scaling.
This method, called BoxBPR, modifies BPR by imposing explicit lower and upper bounds (a "box") on the score difference between positive and negative pairs to simultaneously prevent over-separation and ensure sufficient distinction between positives and negatives. These bounds are pair-specific and dynamically updated during training. In practice, this is implemented by changing the BPR loss (which is a soft differentiable version of hte number of misclassified pairs) into a new objective function representing a soft differentiable version of the two constraints (upper- and lower-bounds). It has the advantage to offer minimal computational overhead.

**Strengths:**

* The paper is generally clear (Except second half of Section 4),  well-structured and offering nice visual illustrations.
* The proposed method is easy to implement and can be integrated easily into any BPR-based framework.
* The method performs robustly across datasets and architectures, with modest yet steady improvement.

**Weaknesses:**

* The theoretical devopments of Section 4 - which constitute the theoretical basis of the method - are not convincing at all. For the lower bound, the argument is also valid for the orginial BPR method (with \alpha =0) for which we have $\hat{r}_{ui} \ge s_u^{(K)}$. So, from this aspect, BoxBPR has virtually no extra advantage wrt BPR. For the upper bound, it seems that the advantage is limited to gradient stabilization (and not aligment with top-K metrics). However, hard negative examples are characterized by the fact that $\Delta_{uij}$ is not so big (otherwise it would be an "easy" negartive) and so, for them, the gradient will not vanish any way.
The whole section need to be much more rigourous  Also remember that these are soft bounds. So, there is strictly speaking no formal guarantees.
* There is no guidelines for the selection of the \alpha and \beta hyper-parameters. There is no probabilistic motivation and these hyperparameters seem to be tuned empirically.
* No Statistical Significance Testing in the experimental part.
* The experimental part is still limited to relatively small scale datasets. Large-scale datasets, with dozens of millions of users/items/interactions exist, such as Amazon Review, Yambda‑5B, HotelRec, ... and testing the BOX-BPR method on these datasets would certainly be valuable.
* As interesting baselines, there are more modern approaches in metrics learning and constrastive learning that already use margins and bounds (ex:https://arxiv.org/html/2307.04047v2, https://arxiv.org/html/2507.14828v1). IT would have been interesting to compare the proposed approach with these more recent approaches.
* BoxBPR remains a pairwise approach. And it is well known that listwise approaches (eg LambdaMart) are often more effective, especially for improving ranking metrics such as TOP-K and nDCG. Once again,  it would have been interesting to compare the proposed approach with these more listwise approaches.

**Questions:**

* Could you explain why, in some situations, there is  some kind of score overlapping between positive and negative items, which is the opposite of excessive separation. Could we predict, depending on the dataset or other factors, whether case 1 or case 2 of Figure 1 will happen?

---

### Official Review · Reviewer_B7UZ · 2025-10-30

**Soundness:** 3
**Presentation:** 3
**Contribution:** 2
**Rating:** 4
**Confidence:** 4

**Summary:**

This paper proposes BoxBPR, a variant of Bayesian Personalized Ranking (BPR) that introduces explicit lower and upper bounds on the pairwise score difference between positive and negative items.

The idea is to constrain the score difference \Delta_{uij} = \hat r_{ui} - \hat r_{uj} within a “box” defined by adaptive parameters \alpha \hat r_{ui} \le \Delta_{uij} \le \beta \hat r_{uj}, thereby preventing excessive score separation or clustering.

The authors provide a probabilistic formulation of the bounded objective, show theoretical connections to Top-20 metrics (HR@20, NDCG@20), and evaluate on three public datasets with consistent improvements over BPR-based baselines.

**Strengths:**

- The motivation is clear. The authors identify real weaknesses of BPR, including score explosion, clustering, and AUC–Top-K mismatch.
- The formulation is simple and general. The method can be easily plugged into existing BPR-based recommenders.

**Weaknesses:**

- There is a lack of theoretical analysis of generalization or loss of superiority. The paper does not analyze how bounding affects model bias and variance, the optimization landscape, or sample complexity.
- Hyperparameter tuning is not well explained. Parameters α\alphaα and β\betaβ are crucial, but no joint optimization procedure or sensitivity interaction is discussed.
- The evaluation scope is narrow. Metrics focus only on the Top-20. Potential trade-offs with AUC, Top-5, and Top-50 metrics are not explored. Since bounding reduces score spread, it may decrease AUC even if Top-K improves. This possibility is untested.
- There is a lack of quantitative case analysis. The paper qualitatively visualizes score distributions but provides no quantitative comparison, such as the variance of score gaps.

**Questions:**

Could you extend the case figure into actual histograms or empirical distributions of score differences?

---

### Official Review · Reviewer_j8n2 · 2025-10-31

**Soundness:** 3
**Presentation:** 3
**Contribution:** 3
**Rating:** 4
**Confidence:** 3

**Summary:**

The paper proposes BoxBPR, a drop-in replacement for the BPR loss that bounds the pairwise score gap between a user’s positive and negative items with a relative lower bound and a relative upper bound. Concretely, it replaces the single BPR term with two sigmoid terms that enforce both constraints and add standard L2 regularization. The authors argue this prevents over-separation and score clustering, better aligns training with top-K objectives, and stabilizes gradients near the hard-negative frontier.

**Strengths:**

1. The paper diagnoses over-separation and clustering under vanilla BPR and visualizes the effect on score distributions.

2. Box constraints integrate directly with common CF backbones (MF, LightGCN, GCL variants) without architecture changes.

3. The lower-bound sufficient condition and the upper-bound gradient analysis align with HR/NDCG goals better than AUC-driven BPR.

**Weaknesses:**

1. The method resembles a margin-based pairwise loss with relative margins. The paper argues variants focus on sampling or extra signals, but it would benefit from a tighter comparison to margin/hinge or temperature-scaled BPR and listwise objectives to isolate what the box adds beyond using a (relative) margin.

2. Only K=20 is reported; datasets are standard but modest; no statistical tests; and stronger baselines (listwise/top-K oriented and margin-ranking) are missing.

3. For backbones with additional losses, improvements shrink, suggesting interactions. More analysis on how BoxBPR co-optimizes (e.g., with contrastive terms) would be useful.

**Questions:**

1. Did you compare against fixed-margin pairwise losses (hinge or margin-ranking) and temperature-scaled BPR to control score scale?

2. Since constraints scale with r̂_ui/r̂_uj, how does BoxBPR behave under backbones with different score ranges?

3. Could the idea extend to a listwise box to more directly optimize NDCG?

---

### Note · Authors · 2025-11-13

I have read and agree with the venue's withdrawal policy on behalf of myself and my co-authors.